# Application of Targeted Optical Coherence Tomography in Oral Cancer: A Cross-Sectional Preliminary Study

**DOI:** 10.3390/diagnostics14192247

**Published:** 2024-10-09

**Authors:** Vera Panzarella, Fortunato Buttacavoli, Vito Rodolico, Laura Maniscalco, Alberto Firenze, Viviana De Caro, Rodolfo Mauceri, Simona E. Rombo, Giuseppina Campisi

**Affiliations:** 1Department of Precision Medicine in Medical, Surgical and Critical Care (Me.Pre.C.C.), University of Palermo, 90127 Palermo, Italy; vera.panzarella@unipa.it (V.P.); rodolfo.mauceri@unipa.it (R.M.); 2Center for Sustainability and Ecological Transition (CSTE), University of Palermo, 90127 Palermo, Italy; alberto.firenze@unipa.it (A.F.); simonaester.rombo@unipa.it (S.E.R.); 3Unit of Oral Medicine and Dentistry for Fragile Patients, Department of Rehabilitation, Fragility, and Continuity of Care, University Hospital “Policlinico Paolo Giaccone” in Palermo, 90127 Palermo, Italy; giuseppina.campisi@policlinico.pa.it; 4Department of Health Promotion, Mother and Child Care, Internal Medicine and Medical Specialties (ProMISE), University of Palermo, 90127 Palermo, Italy; vito.rodolico@unipa.it (V.R.); laura.maniscalco04@unipa.it (L.M.); 5Department of Biological, Chemical and Pharmaceutical Sciences and Technologies, University of Palermo, 90123 Palermo, Italy; viviana.decaro@unipa.it; 6Department of Mathematics and Computer Science (DMeI), University of Palermo, 90127 Palermo, Italy; 7Department of Biomedicine, Neurosciences and advanced Diagnostics (BIND), University of Palermo, 90127 Palermo, Italy

**Keywords:** optical coherence tomography, oral cancer, squamous cell carcinoma of head and neck, oral potentially malignant disorders, precancerous conditions, oral biopsy

## Abstract

Background/Objectives: The diagnosis of oral potentially malignant disorders (OPMDs) and oral squamous cell carcinoma (OSCC) represent a significant challenge in oral medicine. Optical coherence tomography (OCT) shows promise for evaluating oral tissue microstructure but lacks standardized diagnostic protocols tailored to the structural variability and lesions of oral mucosa. Methods: This cross-sectional observational study aims to evaluate the diagnostic accuracy of targeted biopsy-based and site-coded OCT protocols for common OPMDs and OSCC. Adult patients clinically diagnosed with OPMDs, including oral leukoplakia (OL), oral lichen planus (OLP), and OSCC were enrolled. Clinical and OCT evaluation before and after punch scalpel-site registration preceding diagnostic biopsy on the target site was performed. Blinded observers analyzed the OCT scans for OCT-based diagnoses. Sensitivity, specificity, and diagnostic accuracy for OCT evaluations before and after punch scalpel-site registration were statistically compared with histological findings. Results: A dataset of 2520 OCT scans and 210 selected images from 21 patients was obtained. Sensitivity and specificity post-target site registration were high for OSCC (98.57%, 100.00%), OL (98.57%, 98.57%), and OLP (97.14%, 98.57%). The positive predictive values ranged from 97.14% to 100.00%, while negative predictive values ranged from 98.57% to 99.29%. Inter-observer agreements were strong for OSCC (0.84) and moderate for OL (0.54) and OLP (0.47–0.49). Targeted OCT scans significantly improved diagnostic accuracy for all conditions (*p* < 0.001). Conclusions: This preliminary study supports using site-targeted OCT scans followed by a site-targeted punch biopsy, enhancing precision in oral diagnostics. This approach is foundational for developing pioneering automated algorithms guiding oral cancer and pre-cancer diagnosis via OCT imaging.

## 1. Introduction

Oral carcinogenesis encompasses a multifaceted, multistage process leading to malignant transformation of the normal squamous cells of the oral mucosa. This complex development, influenced by several risk factors [1], is a protracted phenomenon that spans years. During this progression, oral potentially malignant disorders (OPMDs) usually emerge, comprising a diverse spectrum of lesions, each with varying temporal and localized transformation risks, acting as precursors to oral squamous cell carcinoma (OSCC). The critical potential of OPMDs to advance to OSCC underscores the urgency for timely detection and intervention.

The possible adoption by oral health practitioners of a proactive approach to early identification and managing these conditions can reduce the burden of oral cancer, fostering a comprehensive and personalized preventive protocol.

In clinical settings, the diagnosis of oral lesions primarily relies on visual inspection, followed by biopsy and histopathological examination, which remains the gold standard for definitive diagnosis. Additional diagnostic modalities include toluidine blue staining, brush biopsy with cytological analysis, and fluorescence visualization (e.g., VELscope) that can help in identifying suspicious areas [2,3]. Advanced imaging techniques, such as high-resolution micro-endoscopy (HRME) and narrow-band imaging (NBI), offer real-time visualization of mucosal abnormalities but are limited by operator dependency and variable specificity [4]. Cross-sectional imaging methods like MRI and CT scans are used for deeper tissue assessment but lack the resolution to discern superficial epithelial changes crucial for early-stage diagnosis. Despite the availability of several oral screening devices [2,3,4,5], the clinical heterogeneity and overlapping features of these disorders pose significant challenges in accurate differentiation between ‘abiding’ OPMDs and those with dysplasia already in progress.

In this scenario, the diagnostic delay in oral cancer remains paradoxically high, with a late diagnosis rate still high for an anatomical region that is among the easiest to inspect. Consequently, the mortality rate of OSCC patients has remained unchanged over the last 20 years, despite the innovative screening and therapeutic techniques available today [6,7]. Non-invasive, standardized, and easy-to-use strategies remain the key method to promote early diagnosis and improve the prognosis of OPMDs and OSCC. In oral oncology, optical techniques have been used to non-invasively provide information regarding the biological tissue changes in optical characteristics that might result in improved life-saving outcomes for oral cancer patients, from screening and staging to follow-up [8,9].

Among these, Optical Coherence Tomography (OCT) is an emerging technology enabling cross-sectional imaging of biological tissues [10,11,12], permitting a non-invasive evaluation of ultrastructural characteristics of mucosal organizations. The potential validity of OCT in oral carcinogenesis has been widely investigated since several ex vivo and in vivo studies have compared the OCT images of normal and pathological lesions to histological results [13,14,15,16,17]. OCT proved a potential diagnostic indicator of progressive tissue transformation scans from normal epithelium to early invasive carcinoma in the oral cavity [11,18,19,20,21,22]. Especially in the artificial intelligence (AI) era, these innovative potentials have also been considered for the development of automated diagnostic-based-OCT algorithms, to improve optical interpretation of oral carcinogenesis [23,24,25,26,27,28].

However, no accurate automated diagnostic tool can be employed without a standardized protocol for the selection and purchase of clinical, digital, optical, and histological images of the oral lesions, which is currently needed, particularly concerning the use of OCT on early oral carcinogenesis [10,29,30,31]. The lack of procedural standardization and interoperability intervenes in all the operational sequences involved in the development of AI models applied to oral cancer diagnosis, starting from the clinical, histological, and OCT-guided evaluation of the suspected lesion. Moreover, in general, to date, there is no validated operational protocol for the correct OCT interpretations of OPMDs and OSCC, which respects their morphological/chromatic heterogeneity (i.e., ulcerative, hyperkeratotic, vegetating lesions) and variable microstructure of the mucosa in several oral sites. This places very strong limits on the possibility of having correct and reproducible interpretations of these lesions, using OCT, making plans for its use for the non-invasive monitoring of OPMD and the early diagnosis of OSCC useless. Particularly, standardized OCT collection of images is essential to ensure that the datasets obtained are comprehensive, reliable, and accurately representative of the targeted pathologies. This entails meticulous attention to sampling technique, and clinical, optical, and histological image labelling, to improve the overall quality and reproducibility of oral carcinogenesis OCT-based interpretation.

This study introduces the first phase of a multistep project aiming to offer a standardized OCT diagnostic algorithm for the most common OPMDs, comprising oral leukoplakia (OL), oral lichen planus (OLP), and OSCC. Particularly, this preliminary cross-sectional investigation aims to validate a clinical protocol based on the use of structured OCT digital lesion patterns. Through the application of these models to OCT scans of targeted biopsy sites, the aim is to enhance the diagnostic precision of OCT for early oral cancer and precursor oral lesions compared to traditional histopathological examination.

To the best of our knowledge, it is the first time that a standardized clinical, digital-OCT and histological design integrated model for the evaluation of OL, OLP and OSCC, to support the development of targeted diagnostic automatized optical algorithms, has been proposed in the literature.

## 2. Materials and Methods

The study protocol conformed to the ethical guidelines of the 1964 Declaration of Helsinki and its later amendments or comparable ethical standards. It was also approved by the Institutional Review Board of University Hospital “Policlinico Paolo Giaccone” in Palermo (Italy), approval number 11/2016. Adherence to research reporting guidelines, specifically STROCSS for cross-sectional studies, ensures comprehensive and transparent reporting of our research findings [32].

### 2.1. Sample Selection

All participants, after providing written informed consent, were consecutively recruited at the Oral Medicine Unit of the University Hospital “Policlinico Paolo Giaccone” in Palermo (Italy), between January and March 2024.

The eligibility criteria were the following:Age ≥ 18 years;Ability to provide informed consent;Clinical diagnosis strongly suggestive of OPMDs and OSCC, according to WHO classification and consensus for oral cancer and head and neck [33,34].

Patients were screened through a comprehensive conventional oral examination (COE) [35] including OCT scans and biopsy, to ensure a comprehensive understanding of the clinical presentation of eligible oral lesions, providing essential contextual information for the subsequent diagnostic procedures.

A digital photographic set per lesion was made to record the site of future evaluation, by OCT and histology, of the lesions. All photographs were taken using a Nikon D7200 Camera, with a Nikon AF-S DX 105 mm F2.8G Lens and Nikon R1C1 dual flash (Nikon Corporation, Tokyo, Japan).

Demographic data on gender and age were collected. Each lesion was site-coded applying the 2021 NIH/SEER ICD-0-3.2 topographical classification codes (from C02.0 to C02.2 for the mobile tongue, C03.0 and C03.1 for the upper and lower gum, respectively, and C06.0 for cheek mucosa, buccal mucosa, and internal cheek) [36,37].

### 2.2. Phase 1: OCT Evaluation Pre-Target Site Registration

For this study, we used the device OCT SS-OCT VivoSight^®^, Michelson Diagnostics Ltd., version 2.0, Orpington, Kent, UK. The system type is a Swept-source Fourier-Domain OCT. The light source of the device is a Santec HSL-2000-12 wide sweep laser with a central wavelength of 1305 ± 15 nm and a frequency sweep range of ≥150 nm. The axial optical resolution in tissues is <10 μm and the lateral resolution is <7.5 μm, with a maximum scan width of 6 mm × 6 mm to a focal depth of ≈2 mm.

The scan obtained was of the “EnFace” type with a default width set to 6 mm with 120 slices, corresponding to a slicing step of 50 μm, for a total scan duration of 12 s.

The preliminary OCT assessment was conducted by the same oral expert (G.C.) on the most clinically suggestive area for each lesion, taking into consideration standardized and histologically compared OCT patterns of OL, OLP and OSCC, validated in the previously published investigations [19,38,39,40,41,42,43,44,45]. In particular, these studies focus on OL, OLP and OSCC, with OCT parameters/patterns evaluating the various mucosal layers (Keratinized Layer (KL), Stratified Epithelial Layer (SEL), Basement Membrane (BM), and Lamina Propria (LP)), compared to healthy mucosa, as detailed in Figure 1.

For each recruited lesion, 120 OCT scans were obtained. From this set, the 10 most representative scans were meticulously selected, employing rigorous criteria grounded in image definition. Preference was given to scans exhibiting optimal visualization of tissue stratification, clear delineation, and pronounced contrast among epithelial components, selecting OCT images that showcased discernible patterns in the most recognizable manner possible.

### 2.3. Phase 2: OCT Evaluation Post-Target Site Registration

After the OCT preliminary evaluation (Figure 2a–f), the target registration of the most representative site for biopsy was performed (Figure 2g,i,k). Specifically, following topical surface anesthesia with lidocaine 25 mg/g + prilocaine 25 mg/g in a cream formulation, the site was marked by the circular blade of a disposable punch biopsy scalpel, 6 mm in diameter (KAI medical, Gifu, Japan), deepened into the supra-epithelial layers, to avoid bleeding and discomfort (≈0.5 mm). Intra-oral photographs of targeted sites were recorded. On this marked site, the same operator (G.C.) carried out a secondary OCT scan session, as previously detailed, obtaining the 10 most representative OCT scans for each lesion, based on the same selected OCT pattern (Figure 2h,j,l).

### 2.4. Phase 3: Targeted Biopsy and Histological Evaluation

After the OCT evaluation steps, the targeted biopsy, with the same punch biopsy scalpel used in the preliminary step, and in the same pre-registered site, was finalized for each lesion, to provide histological confirmation. The surgical margins of the tissue samples were oriented by sutures and photographed to record the preserved orientation equal to the oral in vivo localization.

A comprehensive histological examination, also including the search for epithelial dysplasia, was performed on all collected samples to definitively confirm the initial clinical suspicions. To ensure precise correspondence between the histological analysis and the OCT scans, for each biopsy specimen, while maintaining the spatial orientation, a surgical marker was employed to draw a line across the diametral line. The biopsy specimens were subjected to routine processing, involving fixation in a 10% formalin solution followed by embedding in paraffin. These formalin-fixed, paraffin-embedded (FFPE) samples were then sectioned to a thickness of 5 µm, specifically oriented to correspond with the OCT imaging. These sections were subsequently stained using standard hematoxylin and eosin (H&E) staining techniques and meticulously examined to validate and establish the final diagnosis. The pathologist (VR), responsible for the dissection of the specimens, undertook an independent and blinded evaluation of the histological images. This approach ensured an impartial assessment, free from any influence stemming from the clinical or OCT-based diagnoses.

### 2.5. Phase 4: Blinded Pre- and Post-Site Registration OCT Inter-Comparison to Histological Diagnosis

To strengthen the diagnostic assessment, two distinct OCT examiners (V.P. and F.B.) independently evaluated the OCT images collected pre- and post-site registration. Their OCT evaluations were blinded to both the clinical diagnosis and the histopathological findings. These examiners were assigned the responsibility of discerning structural alterations, employing the same OCT patterns used in the previous steps. The proposed OCT-based diagnoses were then compared to the confirmatory histopathologic diagnoses by a pathologist (V.R.) for both pre- and post-site registration OCT scanning sessions (Figure 2m–o).

To ensure robustness and consistency in the measurements, a secondary scoring round was conducted after a one-month interval, allowing for the evaluation of intra-observer agreement concerning the two separate OCT scan sessions. Throughout this process, a crucial measure was taken to prevent any bias: OCT images were intentionally randomized to guarantee that the initial recording did not exert any influence on the subsequent evaluations. This rigorous methodology was meticulously employed to ensure the accuracy, reliability, and integrity of the diagnostic comparisons between OCT and histological data.

### 2.6. Statistical Analysis

The continuous variables were summarized as mean and standard deviation, while categorical variables were analyzed as counts and percentages. Sensitivity, specificity, and positive and negative predicted values were computed for OSCC, OL, and OLP on preliminary OCT images and OCT after punch-targeted tissue. 95% confidence intervals (95% CI) for sensitivity and specificity were obtained with the exact Clopper-Pearson method. At the same time, 95% CI for predictive values was computed as suggested by Mercaldo et al., 2007, except in the case of 0 or 100% where the Clopper-Pearson method was used [46]. Cohen’s kappa was used to assess the two observers’ agreement on preliminary OCT images and OCT after punch-targeted tissue. Furthermore, the McNemar test for paired data was applied to assess, for each observer, whether the differences between pre- and post-site target registration OCT-based diagnosis affect the identification of the specific disease. Statistical analysis was performed with MedCalc, and an alpha value of 0.05 was considered statistically significant.

## 3. Results

According to the 2021 NIH/SEER ICD-O-3.2 topographical classification, 21 suspected lesions were recruited: seven homogeneous OL (three from the buccal mucosa (C06.0, mean age: 68.7 years; SD: 6.65), four from the ventral tongue (C02.2, mean age: 69.5 years; SD: 6.80)); seven reticular/plaque OLP (four from the buccal mucosa (C06.0, mean age: 64.5 years; SD: 7.89), three from the dorsal tongue (C02.0, mean age: 61.3 years; SD: 1.70)); seven OSCC (three from the anterior inferior alveolar mucosa (C03.1, mean age: 70.7 years; SD: 1.70), four from the lateral tongue border (C02.1, mean age: 75.0 years; SD: 0.82)).

For each lesion, a total of 120 OCT scans were acquired in both OCT evaluation sessions (phases 1–2), resulting in a cumulative dataset of 2520 scans per session. After the selection of the 10 most representative OCT images for each lesion, a total of 210 scans per session was finally processed for the next evaluation phases.

In particular, targeted biopsies were performed, and histological diagnosis was confirmed for all 21 recruited lesions: seven OL, seven OLP and seven OSCC (phase 3). For OL and OLP, no dysplasia was detected.

The diagnostic accuracy of pre- and post-site registration OCT-based diagnoses for both operators was evaluated (phase 4), showing increased sensitivity and specificity values with the application of target scanning for all lesions. Notably, post-target scans exhibited a sensitivity of 98.57% for OSCC, with a specificity of 100.00%. For OL, sensitivity reached 98.57% and specificity was 98.57%. Regarding OLP, sensitivity was 97.14% and specificity was 98.57%. Positive predictive values for OSCC were 100.00%, while for OL and OLP they were 97.18% and 97.14%, respectively. Similarly, negative predictive values exceeded 99% for all conditions (Table 1).

The correctness and correspondence between the OCT-based diagnosis proposed by each observer and the histological diagnosis were confirmed by the McNemar test. The percentage of preliminary OCT images, scanned before the target site registration, and interpreted with the correct diagnosis compared to histopathology, reached over 68.6% for both scoring sessions; higher percentages were obtained for OSCC images (82.9%). The percentages of targeted-OCT images (scanned on the marked site) and interpreted as correct compared to the histopathological diagnosis were higher, with percentages reaching over 97.1%; higher percentages were obtained for OSCC and OL scans (98.6%), slightly lower for OLP (97.1%) for both OCT sessions. The increase in the values of diagnoses corresponding to histopathology, based on OCT scans made on punch-targeted tissue, was statistically significant both for OPMDs (*p* < 0.001 both for OL and OLP) and for OSCC (*p* = 0.001) (Table 2).

The Cohen’s kappa value for identification of OSCC inter-observer agreement was 0.84 (95% CI = 0.76–0.92), representing very good agreement. It was equal to 0.54 (95% CI = 0.42–0.66) for OL and 0.47 (95% CI = 0.34–0.59) for OLP in pre-target, representing for both a moderate agreement. In the post-target phase, the inter-observer agreement was 0.84 (95% CI = 0.76–0.92), representing very good agreement for OSCC, 0.54 (95% CI = 0.42–0.66) for OL and 0.49 (95% CI = 0.37–0.61) for OLP, expressing for both a moderate agreement.

## 4. Discussion

The findings of this cross-sectional study suggest that the use of standardized OCT patterns, particularly defined for OSCC, OL and OLP, and standardized data acquisition procedures in the context of different morphology of oral mucosa and lesions, could enhance the accuracy of OCT in oral cancer diagnosis.

An important lack of uniformity still exists in the literature regarding the OCT in vivo preliminary interpretations and the concordance of clinical, optical, and histological exploration, especially referring to OSCC and OPMDs and OCT procedures (Table 3).

Lesions selection and histological diagnosis appear extremely heterogeneous, especially in studies where the clinical apparency is not well defined; while most studies define the lesions to be included, selecting on appearance (white, red, or white and red lesions) as clinically assimilable to OPMDs [19,38,39,40,42,43,44,45]. Also, the distribution of lesions within the oral cavity exhibits notable variability. All studies process OCT evaluations of the lesions by comparison with healthy mucosa, and most base this assessment on the same anatomical site [19,38,39,40,42,43,44,45]. In our previous investigation and recent studies of Gambino et al., OSCC lesions were compared systematically to the healthy normal mucosa of the same site, revealing the importance of site localization for OCT diagnostic accuracy [19,39,45].

In this study, the integration of in vivo OCT assessments, histologically based, with a site-targeted punch biopsy technique, coupled with the use of standardized diagnostic patterns, distinguishes this research from existing studies. Primarily, the choice to select homogeneous lesions (i.e., plaque for OL, and reticular/plaque for OLP), reflects the need to optically characterize lesions that clinically may present with sometimes overlapping and non-unique appearances. Moreover, the adoption of standardized oral site coding, based on the 2021 NIH/SEER ICD-0-3.2 topographical classification codes, offers precise location-specific OCT analysis of oral lesions and enables a comparison of results across studies, fostering scientific collaboration and data sharing. This may enhance coordination in the management of patients with suspected oral lesions and contribute to the reliability and validity of optical research findings, particularly essential for OPMDs and OSCC early diagnosis.

Furthermore, the use of uniform and validated OCT patterns for OL, OLP and OSCC discrimination guarantees the competitive development of non-invasive, rapid, reproducible, simple-to-use diagnostic algorithms that can be useful not only to oral medicine specialists.

These patterns guide both non-invasive preliminary evaluations and targeted biopsies, ensuring precise alignment between OCT-based hypotheses and histological findings. In this regard, to improve the optical evaluation of the lesions selected for our sample model, we defined a topographic correspondence system that was functional both for OCT validation and for the choice of the most appropriate biopsy site. The punch-based target site co-registration procedure enhances the diagnostic accuracy of OCT compared to histopathological reference: sensitivity and specificity for OCT scans post-target site registration were 98.57 and 100.00 for OSCC, 98.57 and 98.57 for OL, 97.14 and 98.57 for OLP, respectively, for both observers. The substantial increase in the diagnostic accuracy between pre- and post-target site registration OCT-based diagnoses corresponding to histopathology demonstrated a statistical significance: both for OPMDs (*p* < 0.001 for both OL and OLP) and for OSCC (*p* = 0.001).

The choice to perform OCT scans both before and after punch site registration contributes to the uniqueness of the study design. Moreover, the circular punch employed in this technique boasts a diameter of 6 mm, precisely matching the diameter of the scan produced by the utilized OCT probe. As a result, tissue marking with the punch enables alignment not only in terms of location but also in terms of size between the scanned target area and the histologically analyzed tissue sample.

The highest diagnostic accuracy values achieved for OSCC through post-site registration assessments were attributed to the greater OCT consistency of patterns used for this disease compared to the increased variability associated with the patterns of the OPMDs under investigation (OL, OLP). The complete absence of well-defined epithelial stratification, together with the presence of ‘icicle-like’ structures, suggestive of tumor progression/invasion, render the OCT patterns for OSCC more uniform and straightforward to interpret [19]. On the contrary, the report of slightly lower diagnostic accuracy for OLP, despite the morphological homogeneity of the selected lesions (i.e., hyperkeratotic/reticular) could plausibly be attributed, in our assessment, to this intrinsic heterogeneity of pathology, depending on the inflammatory portion underlying the examined layer, underscoring the importance of considering the complexity of OLP tissues when evaluating OCT results. Overall, the contribution of multiple clinical/optical/histological blinded approaches adds consistency and objectivity to this study design and more reliability to OCT-based diagnoses.

Regarding imaging depth, OCT typically provides effective visualization up to approximately 2 mm within the tissue. While this depth is generally sufficient for assessing superficial epithelial changes and all mucosal structures, it may limit the evaluation of deeper tissue layers beneath the lamina propria. For lesions or conditions involving deeper tissue changes, additional imaging techniques or diagnostic methods may be necessary. However, for the purposes of our study, the depth achieved was adequate to analyze and distinguish between various oral lesions, particularly those affecting the epithelial and superficial subepithelial layers. This study focused on these accessible layers to validate standardized OCT patterns for early diagnosis and monitoring, acknowledging that deeper tissue analysis may require complementary diagnostic approaches to fully assess lesions extending beyond this depth.

The findings of this study could have significant implications for clinical practice. The high diagnostic accuracy achieved with the site-targeted punch biopsy-based OCT technique, especially in distinguishing OSCC from OPMD, suggests that this method could be pivotal in reducing diagnostic delays and improving treatment outcomes. Moreover, the implementation of standardized OCT patterns and site-specific coding can enhance the reproducibility and consistency of diagnoses across different clinical settings, potentially leading to more widespread adoption of OCT in routine oral cancer screenings. Furthermore, the ability to accurately target biopsy sites based on OCT findings can minimize the need for multiple invasive procedures, reducing patient discomfort and the risk of sampling errors. This not only streamlines the diagnostic process but also supports more personalized treatment planning at the earliest possible stage.

As the integration of Artificial Intelligence (AI) with OCT continues to evolve, these standardization protocols could serve as a foundational framework for the development of more sophisticated, automated diagnostic algorithms, currently missing from the literature.

Kim D. H. et al. (2022) conducted a comprehensive systematic review and meta-analysis of OCT efficacy in oral oncology, highlighting the diagnostic potential of AI and automated algorithms but not noting a dearth of studies emphasizing the precise spatial overlap between OCT-scanned areas and subsequently biopsied regions for diagnostic reference [23]. This oversight introduces a notable limitation in correlating OCT findings directly with histological outcomes. Similarly, Kim J-S (2022), in their exploration of the integration of OCT and AI for discriminating oral cancerous lesions from normal mucosa, included an analysis of four relevant studies [47]. Notably, despite the confirmation of diagnoses through histological examination in all four studies, none explicitly addressed the precise spatial overlap between the biopsy-sampled areas and those identified through OCT scans [24,25,29,48].

The need for spatial concordance between OCT-scanned regions and the corresponding biopsy sites is essential to reduce the qualitative interpretative complexity of oral lesions in the early learning phase of any applied computational OCT system [47,49]. Our study could contribute to filling this gap by deliberately incorporating a site-targeted punch biopsy-based technique in new automatized OCT-supported clinical algorithms, especially designed for oral carcinogenesis diagnosis.

The limitations of this study underscore several constraints that must be considered when interpreting its findings. As a preliminary phase of multi-project research, this preliminary investigation was designed as unicentric on a relatively small sample size. Moreover, the selection of only specific anterior mucosal sites of the mouth was influenced by the limitations of the OCT probe utilized. These choices may introduce the possibility of site-specific biases and limit the broader applicability of the findings. Future studies are currently underway to validate our protocols and applications in multicenter settings, with a larger sample of patients, lesions, and mucosal sites.

## 5. Conclusions

This research study presents an innovative procedure introduced for the diagnosis of oral pre-cancer and cancer, based on OCT, adherent to a strictly standardized protocol. By providing a meticulously standardized dataset, our study lays a solid foundation for refining and advancing automated diagnostic applications, encouraging a virtuous cycle of iterative improvement in optical diagnostic precision. Thus, this work highlights the transformative potential of harmonizing standardized clinical methodologies with future AI technologies in revolutionizing early cancer detection paradigms, aiming to promote advancements in patient care and anticipate further progress in the field, leading to improved patient outcomes and overall survival rates.

## Figures and Tables

**Figure 1 diagnostics-14-02247-f001:**
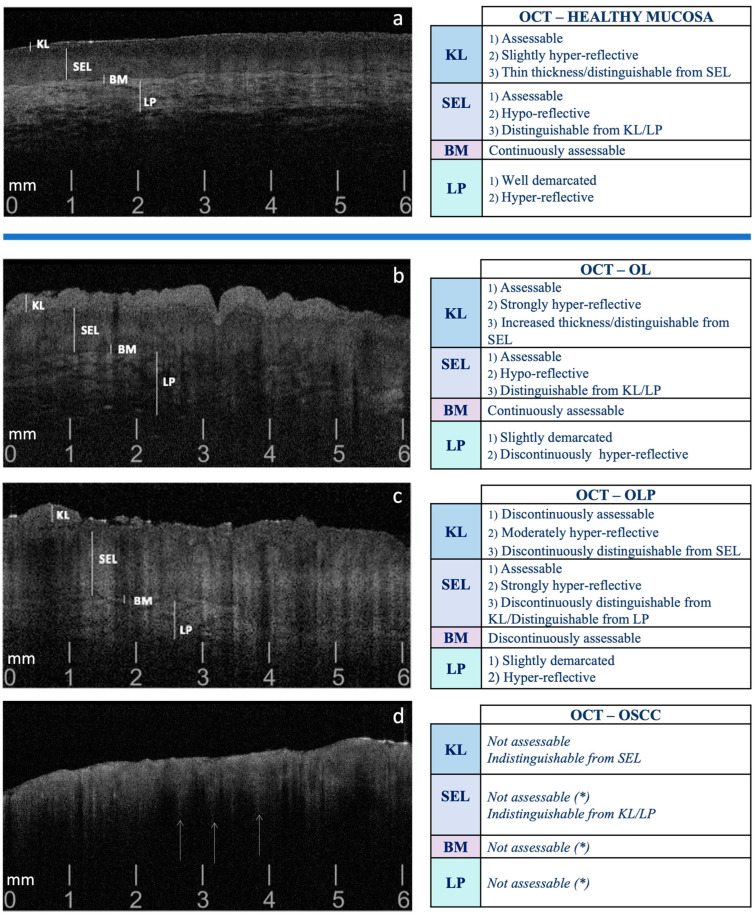
Guiding pattern criteria for the selection of OCT scans, comparing the patterns observed in healthy oral mucosa (**a**) with those in OL (**b**), OLP (**c**), and OSCC (**d**). OL: Oral Leukoplakia; OLP: Oral Lichen Planus; OSCC: Oral Squamous Cell Carcinoma; KL: Keratinized Layer; SEL: Stratified Epithelial Layer; BM: Basement Membrane; LP: Lamina Propria. (*) Presence of ‘icicle-like’ structures: hyper-reflective conical configurations that extend from the superficial cellular layers (SEL) to the deeper ones (BM and LP), commonly reported in OSCC, suggestive of neoplastic intra/sub-epithelial infiltration (indicated by white arrows ↑) [19].

**Figure 2 diagnostics-14-02247-f002:**

Pre- and post-target site registration clinical, and related OCT most representative scans, compared to histopathology confirmatory images of three selected cases of OL, OLP and OSCC (**a**,**b**): pre-site registration clinical image (**a**) and OCT scan (**b**) of a case of homogeneous OL on the left buccal mucosa. (**c**,**d**): pre-site registration clinical image (**c**) and OCT scan (**d**) of a case of reticular OLP on the dorsal surface of the tongue. (**e**,**f**): pre-site registration clinical image (**e**) and OCT scan (**f**) of a case of OSCC on lower anterior alveolar mucosa. (**g**,**h**): post-site registration clinical image (**c**) and OCT scan (**d**) of the previous OL case. (**i**,**j**): post-site registration clinical image (**i**) and OCT scan (**j**) of the previous OLP case. (**k**,**l**): post-site registration clinical image (**k**) and OCT scan (**l**) of the previous OSCC case. (**m**–**o**): histopathology confirmatory images (H&E stain; original magnification ×2.5) of the previous OL, OLP and OSCC cases, ((**m**), (**n**), (**o**) respectively). KL: Keratinized Layer; SEL: Stratified Epithelial Layer; BM: Basement Membrane; LP: Lamina Propria; white arrows (↑): ‘icicle-like’ structures.

**Table 1 diagnostics-14-02247-t001:** Diagnostic accuracy measures for OCT compared to histopathology findings. Pre-target indicates the OCT evaluation pre-target site registration; post-target indicates the OCT evaluation post-target site registration.

			Predictive Value % (95% CI)	
Observer 1	Sensitivity % (95% CI)	Specificity % (95% CI)	Positive	Negative
Pre-target				
OSCC	82.86, (71.97–90.82)	97.86, (93.87–99.56)	95.08, (86.26–98.35)	91.95, (87.21–95.03)
OL	70.00, (57.87–80.38)	85.00, (77.99–90.47)	70.00, (60.45–78.08)	85.00, (79.74–89.08)
OLP	68.57, (56.37–79.15)	77.86, (70.07–84.43)	60.76, (52.21–68.70)	83.21, (77.61–87.63)
Post-target				
OSCC	98.57, (92.30–99.96)	100.00, (97.40–100.00)	100.00, (94.79–100.00)	99.29, (95.24–99.90)
OL	98.57, (92.30–99.96)	98.57, (94.93–99.83)	97.18, (89.70–99.27)	99.28, (95.17–99.90)
OLP	97.14, (90.06–99.65)	98.57, (94.93–99.83)	97.14, (89.56–99.26)	98.57, (94.62–99.63)
Observer 2				
Pre-target				
OSCC	82.86, (71.97–90.82)	98.57, (94.93–99.83)	96.67, (87.94–99.14)	92.00, (87.29–95.06)
OL	70.00, (57.87–80.38)	84.29, (77.18–89.88)	69.01, (59.57–77.10)	84.89, (79.60–89.00)
OLP	68.57, (56.37–79.15)	78.57, (70.84–85.05)	61.54, (52.88–69.52)	83.33, (77.78–87.72)
Post-target				
OSCC	98.57, (92.30–99.96)	100.00, (97.40–100.00)	100.00, (94.79–100.00)	99.29, (95.24–99.90)
OL	98.57, (92.30–99.96)	98.57, (94.93–99.83)	97.18, (89.70–99.27)	99.28, (95.17–99.90)
OLP	97.14, (90.06–99.65)	98.57, (94.93–99.83)	97.14, (89.56–99.26)	98.57, (94.62–99.63)

**Table 2 diagnostics-14-02247-t002:** McNemar test for paired data conducted on OCT evaluation pre- and post-target site registration. Pre-target indicates the OCT evaluation pre-target site registration; post-target indicates the OCT evaluation post-target site registration (* 95% CI was reported as the difference percentage).

Observer 1	Pre-Target *n*. (%)	Post-Target *n*. (%)	Difference	95% CI, % *		*p*-Value
OSCC	58 (82.9)	69 (98.6)	15.71%	7.19	24.24	0.001
OL	49 (70)	69 (98.6)	28.57%	17.99	39.15	<0.001
OLP	48 (68.6)	68 (97.1)	28.57%	17.99	39.15	<0.001
Observer 2						
OSCC	57 (81.4)	69 (98.6)	17.14%	8.31	25.97	<0.001
OL	50 (71.4)	69 (98.6)	27.14%	16.73	37.56	<0.001
OLP	49 (70)	68 (97.1)	27.14%	16.73	37.56	<0.001

**Table 3 diagnostics-14-02247-t003:** Characteristics of the studies investigating in vivo/ex vivo application of OCT for preliminary assessment of OSCC and OPMDs. BM: Basement Membrane; CIS: Carcinoma in situ; EP: Squamous stratified epithelium; EP Re: Reflectivity of epithelial layer; ET: Epithelial Thickness; DG: Desquamative Gingivitis; GVHD: Graft Versus Host Disease; K-micro: Micro-invasive carcinoma; KL: Keratinized Layer; LP: Lamina Propria; LP Re: Reflectivity of lamina propria; MMP: Mucous Membrane Pemphigoid; OL: Oral Leukoplakia; OLP: Oral Lichen Planus; OSCC: Oral Squamous Cell Carcinoma; PVL: Proliferative Verrucous Leukoplakia; PV: Pemphigus Vulgaris; SEL: Stratified Epithelial Layer; SS: Stratified Squamous epithelium.

	Author (Year)	Study Design	*N*. Cases	Clinical Appearance	Pathological Diagnosis	Oral Sites	OCT Patterns Parameters/Evaluation
1	Ridgway (2006) [41]	Case series	41	Not specified benign and malignant lesions	Begnin lesions, OL, CIS, OSCC	Buccal mucosa, Floor of mouth, Gingiva, Hard palate, Lip, Tongue	SS, BM/LP
2	Wilder-Smith (2009) [40]	Preliminary Study	50	Leukoplakia, erythroplakia	Dysplasia (mild, moderate, severe) CIS, OSCC	Tongue, Buccal mucosa, Floor of mouth	SEL, BM, LP
3	Volgger (2012) [42]	Prospective diagnostic trial	100	Leukoplakia, erythroplakia	OSCC, dysplasia, OLP	Buccal mucosa, Floor of mouth, Gingiva, Palate, Lip, Tongue (detailed only for healthy mucosa)	KL, EP, ET, BM, LP
4	Gambino (2020) [38]	Case control study	20	Atrophic-erosive OLP	OLP	Buccal mucosa	EP, BM, LP
5	Panzarella (2021) [43]	Observational study	43	DG	OLP, PV, MMP	Gingiva	SEL, BM, LP
6	Panzarella (2022) [19]	Descriptive pilot Study	30	Suspected OSCC	OSCC	Tongue, Gingiva, Buccal mucosa	KL, SEL, BM, LP
7	Gambino 2022 [45]	Case/control study	50	Non-healing Ulcerations	Traumatic lesions, OSCC	Buccal mucosa, Gingiva, Tongue	EP, BM, LP
8	Gambino (2023) [39]	Case series	11	White, red-white lesions, ulcers	PVL, OLP, OL, GVHD, K-MICRO	Tongue, Gingiva, Buccal mucosa	KL, EP, BM, LP
9	Gruda (2023) [44]	Case series	15	OLP, leukoplakia	OLP, OL, OSCC	Buccal mucosa, Tongue	EP, EP Re, LP Re, BM

## Data Availability

The datasets generated and/or analyzed during the current study are available from the corresponding author on reasonable request.

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
