# Peer review of "Application of Targeted Optical Coherence Tomography in Oral Cancer: A Cross-Sectional Preliminary Study"

_diagnostics, 2024, doi:10.3390/diagnostics14192247_

Round 1

Reviewer 1 Report

Comments and Suggestions for Authors

This manuscript demonstrates the preliminary study for diagnosing the oral potentially malignant disorder and oral squamous cell carcinoma using optical coherence tomography (OCT). The following questions should be addressed and incorporated in the manuscript:  

- What are the current diagnostic techniques in hospital to diagnose the oral tissue? It can be briefly mentioned in the introduction. 

- Figure 1: what is the scale bar on the images to understand the size of different layers in healthy vs non-healthy oral tissue.

-As OCT can image around 2mm of depth within the tissue. Is it important to image beyond this depth for diagnosis?

- In figure 2, along with the histology images of OL, OLP and OSCC, is there any representative histology image of healthy oral tissue to map the different layers on OCT.

-  What is the size of different layers in different condition vs healthy tissue? Average size of different layers observed in all the samples should be mentioned in the results.

- Does the cross-sectional structure of buccal mucosa, ventral and dorsal tongue, anterior inferior alveolar mucosa are different or same considering the healthy vs diseased tissue? Do the size of different layers change with the tissue area selected?

Author Response

Response to Reviewer 1’s Comments - Manuscript ID: diagnostics-3188312

Dear Reviewer 1,

We would like to thank you for your insightful comments on our manuscript titled “Application of Targeted Optical Coherence Tomography in Oral Cancer: A Cross-Sectional Preliminary Study” (Manuscript ID: diagnostics-3188312). We have carefully addressed each of your suggestions, and we believe that these revisions have strengthened our study. Below, please find our responses to your comments:

Comment 1: What are the current diagnostic techniques in hospital to diagnose the oral tissue? It can be briefly mentioned in the introduction.

  • Response 1:Thank you for pointing this out. We added a brief mention (lines 55-64) of the current diagnostic techniques for oral tissue diagnosis in the introduction, including visual inspection, biopsy with histopathological analysis, and other imaging methods.

Comment 2: Figure 1: What is the scale bar on the images to understand the size of different layers in healthy vs non-healthy oral tissue.

  • Response 2: Thank you for highlighting this aspect. The scale of measurement for these structures with OCT varies depending on the specific OCT system and its resolution. Generally, such measurements are reported in tenths of a millimeter. However, due to the considerable diversity among OCT machines and the heterogeneity in their capabilities, precise quantitative measurements can be challenging. Not all OCT systems offer the same level of measurement accuracy, and these measurements are also very influenced by operator variability. In our study, we did not perform direct quantitative measurements of the epithelial layer and lamina propria thickness. Instead, we concentrated on visual pattern analysis using OCT to ensure a consistent and objective interpretation of the OCT images.

In this research, the standard dimensional scale for the OCT images is consistently set to 6 mm for the entire length of the image (i.e., the real dimension of the scanned oral area by OCT probe). The scale bar of 6 mm in Figure 1 was not included specifically to emphasize the focus on visual pattern analysis, as Figure 1 is intended to be illustrative of these patterns. However, the scale remains consistent across all OCT images in the study (Figure 2, for instance, shows the 6 mm scale for both OCT scans and confirmatory histopathology images). As specified in the Materials and Methods section, this 6 mm dimension is also applied to the corresponding histological sections, in line with the study's objectives. We hope this clarification addresses your query.

Comment 3: As OCT can image around 2mm of depth within the tissue. Is it important to image beyond this depth for diagnosis?

  • Response 3:A statement (lines 380-390) has been added to the discussion section noting that while OCT's imaging depth of approximately 2mm is generally sufficient for diagnosing oral lesions analyzed, imaging beyond this depth might be relevant in cases of deeper tissue involvement.

Comment 4: In Figure 2, along with the histology images of OL, OLP, and OSCC, is there any representative histology image of healthy oral tissue to map the different layers on OCT.

  • Response 4:

Thank you for your insightful question. We did not include any histological images of healthy oral tissue in our study primarily for ethical reasons, as obtaining such biopsies from healthy individuals would not be appropriate due to the invasive nature of the procedure.

Instead, Figure 1 serves as a reference, providing a detailed comparison of OCT images for OL, OLP, and OSCC with healthy mucosa. This allows for an understanding of the different layers and their appearances in OCT scans. As per the aims of our study, the reference figure of healthy mucosa enables us to map the various layers and their OCT appearances, offering valuable context for interpreting the variations seen in pathological conditions. Although we did not include a direct histological image of healthy mucosa, this comparison allows for the assessment of structural differences between healthy and pathological tissues.

Comment 5: What is the size of different layers in different conditions vs healthy tissue? Average size of different layers observed in all the samples should be mentioned in the results.

  • Response 5:

Thank you for your valuable input. As mentioned previously, in our study, we did not perform direct measurements of the thickness of the epithelial layer and lamina propria due to limitations inherent in OCT systems and potential variability. While OCT systems typically provide measurements in tenths of a millimeter, the accuracy of these measurements can be affected by the specific technology used and the operator's technique. Given these constraints, we focused on visual pattern analysis to interpret the structural characteristics of the oral mucosa.

Our research emphasizes a comprehensive qualitative assessment of the variations in tissue layers across different pathological conditions compared to healthy tissue. This analysis is thoroughly detailed in the results and discussion sections of the manuscript. We appreciate your understanding of the limitations in our approach and hope the qualitative insights provided contribute meaningfully to the understanding of these tissue variations.

Comment 6: Does the cross-sectional structure of buccal mucosa, ventral and dorsal tongue, anterior inferior alveolar mucosa differ or are they the same considering healthy vs diseased tissue? Do the sizes of different layers change with the tissue area selected?

  • Response 6:Thank you for your thoughtful question. Indeed, the ultrastructural characteristics of healthy oral mucosa, as visualized in OCT scans, vary depending on the anatomical site. Different sites, such as the buccal mucosa, the ventral and dorsal aspects of the tongue, and the anterior inferior alveolar mucosa, each exhibit unique structural features and layer thicknesses. Additionally, the visibility and distinctiveness of these layers can vary significantly across different sites, affecting how their characteristics are visualized and interpreted in OCT scans. This site-specific variation was a key focus in our previous research, where we systematically compared oral cancer-affected tissue with corresponding healthy tissue (https://doi.org/10.3390/cancers14235916). Our study utilized site-specific and coded OCT scans to highlight these differences. These findings underscore the importance of considering anatomical variability when interpreting OCT results and emphasize the need for standardized protocols that account for such site-specific characteristics. We hope this clarifies the variations in tissue structure and visualization observed in different oral sites and the impact of these differences on the diagnostic analysis.

We appreciate your valuable feedback and believe these revisions address all your concerns. Thank you again for your thoughtful review.

Best regards,

Fortunato Buttacavoli

DDS, PhD Student, Postgraduate Specialist in Pediatric Dentistry

Department of Precision Medicine in Medical, Surgical and Critical Care (Me.Pre.C.C.)

University of Palermo, Via Liborio Giuffrè, 5 - 90127

fortunato.buttacavoli@unipa.it

Reviewer 2 Report

Comments and Suggestions for Authors

Dear authors, 

thank you for selecting this topic for your manuscript.

I find it very useful for daily clinical practice and science in general.

In my opinion it should be accepted for publication after minor grammar and spell check

Congrats to the authors and kind regards!

Comments on the Quality of English Language

Only minor grammar and spell check is needed.

Author Response

Response to Reviewer 2’s Comments - Manuscript ID: diagnostics-3188312

Dear Reviewer 2,

We would like to thank you for thoroughly evaluating our manuscript titled “Application of Targeted Optical Coherence Tomography in Oral Cancer: A Cross-Sectional Preliminary Study” (Manuscript ID: diagnostics-3188312). We appreciate your positive feedback and comments.

Comment: Thank you for selecting this topic for your manuscript. I find it very useful for daily clinical practice and science in general. In my opinion, it should be accepted for publication after minor grammar and spell check. Congrats to the authors and kind regards!

  • Response:Thank you very much for your positive feedback and encouraging comments. We are delighted to hear that you find our study useful for clinical practice and science. We appreciate your suggestion regarding grammar and spelling improvements. We ensure that the manuscript has already undergone a thorough review to address these minor issues before resubmission.

Thank you again for your support and kind regards.

Sincerely,

Fortunato Buttacavoli

DDS, PhD Student, Postgraduate Specialist in Pediatric Dentistry

Department of Precision Medicine in Medical, Surgical and Critical Care (Me.Pre.C.C.)

University of Palermo, Via Liborio Giuffrè, 5 - 90127

fortunato.buttacavoli@unipa.it

Reviewer 3 Report

Comments and Suggestions for Authors

This is a preliminary study of the use of site-targeted OCT scans followed by site-targeted punch biopsies in the diagnosis of some oral lesions.

The study is original and in continuity with previous studies carried out by the authors on this theme. 

The article is well written, with a clear message for the reader.

The images are of good quality and well explained.

The discussion is well articulated and the limitations of the study are provided.

My comment concerns the choice of the hyperkeratotic or reticular forms of lichen planus rather than the erosive or atrophic forms, which are more often subject to dysplastic or carcinomatous transformation. 

Author Response

Response to Reviewer 3’s Comments - Manuscript ID: diagnostics-3188312

Dear Reviewer 3,

We would like to thank you for thoroughly evaluating our manuscript titled “Application of Targeted Optical Coherence Tomography in Oral Cancer: A Cross-Sectional Preliminary Study” (Manuscript ID: diagnostics-3188312). We appreciate your positive feedback and comments.

Comment 1: This is a preliminary study of the use of site-targeted OCT scans followed by site-targeted punch biopsies in the diagnosis of some oral lesions. The study is original and in continuity with previous studies carried out by the authors on this theme. The article is well written, with a clear message for the reader. The images are of good quality and well explained. The discussion is well articulated and the limitations of the study are provided. My comment concerns the choice of the hyperkeratotic or reticular forms of lichen planus rather than the erosive or atrophic forms, which are more often subject to dysplastic or carcinomatous transformation.

  • Response 1: Thank you for your positive feedback on our study. We appreciate your recognition of the originality, clarity, and quality of our work.

Regarding your comment on the choice of lichen planus forms, we selected the hyperkeratotic and reticular types rather than the erosive or atrophic forms primarily to achieve a more homogeneous optical characterization. These forms often present clearer, more distinguishable features, which are essential for consistent comparisons and accurate OCT assessments. This approach aligns with our aim to enhance diagnostic accuracy by focusing on lesion types that allow for more reliable and reproducible evaluations.

The standardization of OCT patterns and site-specific coding, including the reticular and plaque forms of oral lichen planus (OLP), is intended to support reproducibility and facilitate data sharing across research initiatives. This is crucial for integrating OCT into clinical practice, particularly for oral cancer screening and diagnosis.

The inflammatory nature of erosive and atrophic forms of OLP introduces significant variability in OCT pattern interpretation, potentially affecting diagnostic consistency. By focusing on the hyperkeratotic and reticular OLP forms in this first phase of our project, we aimed to minimize variability and strengthen the reliability of our findings.

We acknowledge the importance of examining various forms of lichen planus, including erosive and atrophic types, in future studies. A structured protocol for these forms is already underway. Expanding our research to include these variants in larger, multicenter trials will address their potential for dysplastic or carcinomatous transformation and further validate our approach.

Thank you once again for your valuable feedback.

Best regards,

Fortunato Buttacavoli

DDS, PhD Student, Postgraduate Specialist in Pediatric Dentistry

Department of Precision Medicine in Medical, Surgical and Critical Care (Me.Pre.C.C.)

University of Palermo, Via Liborio Giuffrè, 5 - 90127

fortunato.buttacavoli@unipa.it